# Implementation strategies to improve outcomes in patients with established cardiovascular disease in sub-Saharan Africa: A systematic review

Leah A. Sanga[1], Jonathan A. Hudson[2], Alexander D. Perkins[1], Adrianna Murphy[3], Anthony Etyang[4], Pablo Perel[1], Anoop S. V. Shah[1,5]*

1 Department of Non-communicable Disease Epidemiology, London School of Hygiene & Tropical Medicine, London, United Kingdom, 2 Kings College London BHF Centre, School of Cardiovascular and Metabolic Medicine & Sciences, London, United Kingdom, 3 Department of Health Services Research and Policy, London School of Hygiene & Tropical Medicine, London, United Kingdom, 4 Dept of Epidemiology and Demography, KEMRI-Wellcome Trust Research Programme, Kilifi, Kenya, 5 Department of Cardiology, Imperial College NHS Trust, London, United Kingdom

* Anoop.Shah@lshtm.ac.uk

## Abstract

Sub-Saharan Africa (SSA) is experiencing an epidemic of cardiovascular disease (CVD). Despite numerous evidence-based therapies and management guidelines for patients with acute or established CVD, significant gaps persist in their implementation in SSA. This systematic review aims to describe, synthesise and identify key gaps in the implementation strategies of evidence-based approaches that can improve clinical outcomes for patients with acute or established CVD in SSA. We searched four databases for studies that examined the implementation strategies of evidence-based interventions for patients with acute or established CVD in SSA. Studies that did not focus on interventions were excluded. The primary outcome was major adverse cardiovascular events including myocardial infarction, stroke, cardiovascular death or hospitalisation. Secondary outcomes included adherence to treatment, improvement in modifiable risk factors, symptom measures, treatment complications, and psychosocial metrics, particularly those related to quality of life. Nineteen studies met the inclusion criteria (nine evaluated patients with heart failure, three evaluated heart failure or ischaemic heart disease, three evaluated ischaemic heart disease, and four evaluated stroke). Of the 19 studies, 14 were targeted at healthcare recipients, two at healthcare workers and three at the healthcare organisation. The most common interventions evaluated were in the field of cardiac rehabilitation. Only three studies (two evaluating stroke and one heart failure) implemented an intervention in the acute setting with the rest evaluating strategies at discharge or in the ambulatory population. No studies evaluated implementation strategies in hospitalised patients with ischaemic heart disease. This study highlights significant gaps in the implementation of interventions in patients

**Data availability statement:** All relevant data are within the paper and its Supporting Information files.

**Funding:** This work was supported by the Global Health Research National Institute of Health and Care Research (to GHR NIHR134544 to PP and AE). The funders had no role in study design, data collection and analysis, decision to publish, or preparation of the manuscript.

**Competing interests:** The authors have declared that no competing interests exist.

with established cardiovascular disease. Gaps were highlighted in the acute care setting, specifically related to cardiac pathologies and implementation strategies targeting pharmacotherapeutic optimisations. We also highlight a notable lack of studies focusing on effective implementation strategies in primary care facilities and lower-level hospital settings.

## SYSTEMATIC REVIEW REGISTRATION

The protocol was registered in PROSPERO prior to the study implementation (ID: CRD42023465781). The protocol can be accessed at crd.york.ac.uk/PROSPERO/display_record.php?RecordID=465781

## Introduction

Sub-Saharan Africa (SSA) is experiencing an epidemic of cardiovascular disease (CVD) [1]. Stroke, heart failure and ischaemic heart disease, the predominant pathologies constituting CVD, now make up an increasing proportion of acute admissions to hospitals in SSA [2–4]. Despite being nearly a decade younger, short and long-term outcomes following acute cardiovascular events are worse across patients in SSA when compared to high-income countries [5–7]. Individuals with acute or established CVD (stroke, heart failure or ischaemic heart disease) suffer from poorer outcomes and significant economic impact making their effective management a public health priority [8–10]. The NCD Countdown 2030 Health Policy paper identifies the management of patients with acute or established cardiovascular disease as crucial interventions. These are prioritized to achieve the United Nations Sustainable Development Goal target 3.4, aiming to reduce premature mortality from non-communicable diseases by a third [11].

Management of acute and established CVD consists of both pharmacological and non-pharmacological (for example dietary changes, physical activity, lifestyle management and cardiac rehabilitation) approaches to mitigate future cardiovascular morbidity and mortality and maximise the function and quality of life of individuals [12,13]. Despite extensive evidence-based guidance for managing patients with acute or established CVD, significant gaps persist in their implementation in SSA. In particular, prescription of and adherence to proven treatments following cardiovascular events are suboptimal in SSA [5,6,14], and much lower than other regions of the world [15,16].

Although research evaluating strategies to improve implementation of evidence-based interventions for primary prevention of cardiovascular disease in SSA has been increasing, similar research for managing patients with established CVD remains limited [17–19]. This systematic review aims to describe, synthesise, and identify key gaps in the implementation strategies of evidence-based pharmacological and non-pharmacological approaches that improve clinical outcomes for patients with acute or established CVD in SSA.

## Methods

### Data sources and search strategy

The study was reported in accordance with the Preferred Reporting Items for Systematic Reviews and Meta-Analyses (PRISMA) guidelines (S1 Checklist) [20]. We searched MEDLINE, EMBASE, Global Health, and Google Scholar for studies that examined the implementation strategies of evidence-based approaches for patients with acute or established CVD in SSA. We manually searched bibliographic references of key papers. Our search strategy utilised the keywords: "cardiovascular disease", "prevention interventions", and "Sub-Saharan Africa". We included studies of any design, in any language, published from database inception up to the 31st of December 2023. The full search criteria are provided in S1 Text. The protocol was registered in PROSPERO prior to the study implementation (ID: CRD42023465781).

### Eligibility criteria

We included studies reporting on interventions for managing acute or established CVD among adults (>=18 years) in the SSA region. Cardiovascular disease was restricted to the following pathologies: stroke, heart failure or ischaemic heart disease. Clinical trials, non-randomised clinical studies, pre- and post-intervention studies, case-control studies, and cross-sectional studies were eligible for inclusion. We did not impose restrictions based on specific definitions or diagnostic criteria of cardiovascular diseases to ensure inclusion of studies that may not have explicitly detailed their criteria. Studies that did not evaluate implementation of interventions were excluded alongside those studies that solely evaluated interventions targeted at primary prevention of cardiovascular disease.

### Outcome

The primary outcome was major adverse cardiovascular events, which included recurrent myocardial infarction, stroke, cardiovascular death, and hospitalisation due to heart failure or other cardiovascular causes. Given the expected heterogeneity of studies and diverse outcomes, secondary outcomes were also assessed. These included adherence to treatment, improvement in modifiable risk factors, hospitalizations, symptom measures, treatment complications, and psychosocial metrics, particularly those related to quality of life.

### Study selection

Two reviewers (LAS & JAH) independently screened and assessed studies for eligibility based on the inclusion and exclusion criteria. Conflicts were resolved through consensus or consultation with a third reviewer (ASVS). Data, including study characteristics and baseline population details, were extracted by one investigator (LAS) and verified by another (JAH). All studies identified were imported to Rayyan [21], an internet-based program that facilitates collaboration between investigators during the screening and selection of studies to be finally included in the review. Duplicates were removed.

### Evidence synthesis and risk of bias

A narrative synthesis was conducted to provide an overview of the findings from the included studies, focusing on the type and effectiveness of implementation strategies to improve outcomes in patients with CVD in SSA. We categorised the implementation strategies based on the target actor being the healthcare recipient, healthcare worker or the healthcare organisation as defined by the Cochrane Effective Practice and Organisation of Care Group [17,22].

We evaluated the reporting and methodological quality of the included studies using the Evidence project risk of bias tool [23]. We specifically chose this risk of bias tool as it encompassed both randomised and non-randomised intervention studies. This tool awards one point for each of the following eight items: (1) prospective cohort; (2) control or comparison group; (3) pre/post intervention data; (4) random assignment of participants to the intervention; (5) random selection of

subjects for assessment; (6) follow-up rate of 80% or more; (7) comparison groups equivalent on socio-demographic measures; and (8) comparison groups equivalent at baseline on outcome measures.

## Results

### Review of search results

The search yielded 14,435 results, and six additional studies were identified from bibliographic references giving a total of 14,435 studies reviewed. After screening, 34 studies were read in detail to assess eligibility. Of these, 19 met the inclusion criteria and were included in the final analysis (Fig 1 and S1 Table). This included six studies identified via the bibliographic search. We did not find any studies that required translation to English.

### Characteristics of the studies included

**Geographical distribution.** Out of the 19 studies included, one study (5.3%) was international, encompassing three SSA countries (South Africa, Mozambique and Nigeria) [24]. The remaining studies were conducted in single countries as follows: five studies (26.3%) in South Africa [25–29], four (21%) in Nigeria [30–33], two (10.5%) in Ghana [34,35]

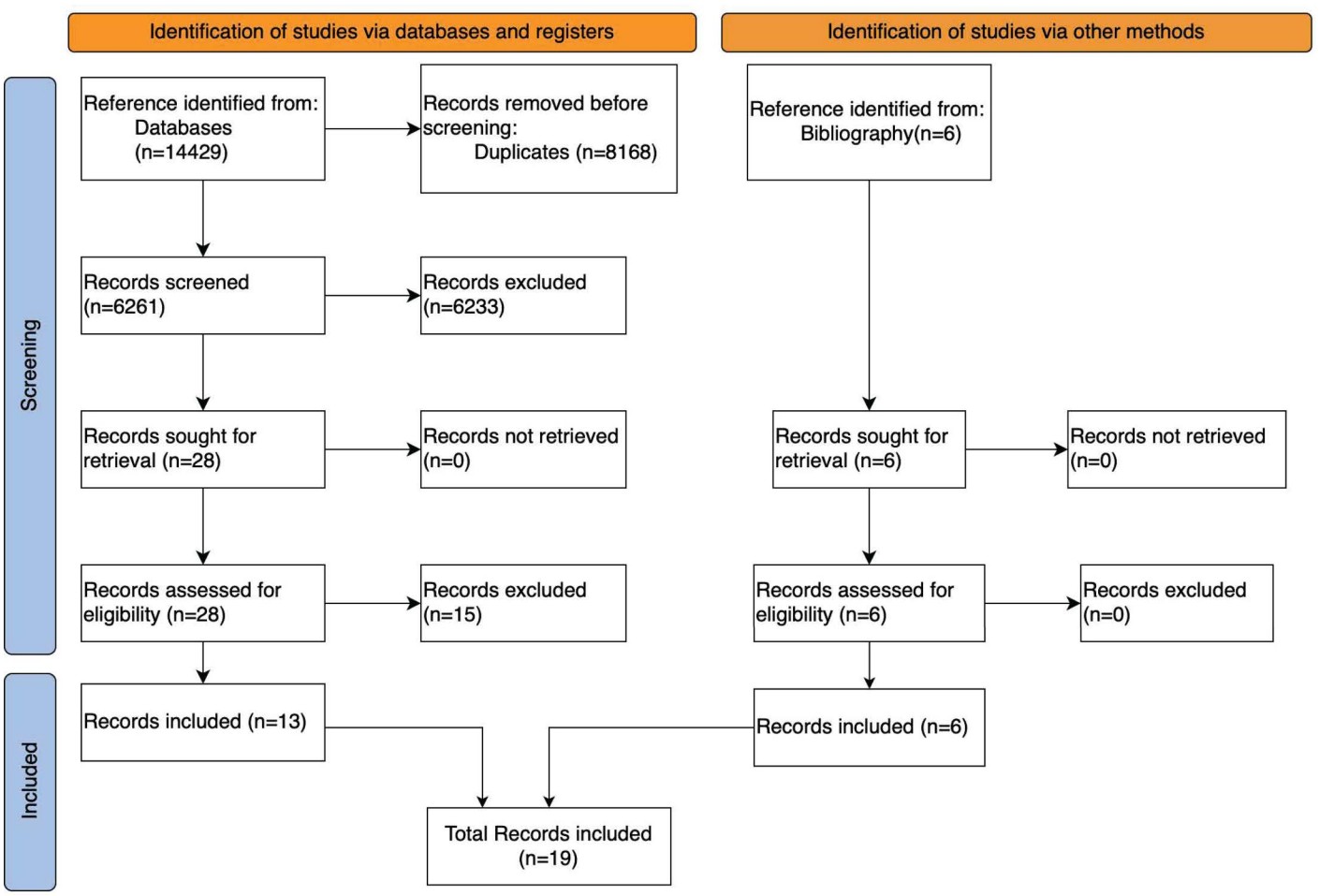

**Fig 1. PRISMA flow chart of studies identified, screened, and included.**

and Ethiopia [36,37], one (5.3%) in Uganda [38], Sudan [39], Rwanda [40], Kenya [41] and Benin [42] (Table 1 & Fig 2). Studies included in this review represented 10 (20%) of the 49 countries in the SSA region.

## Study setting and population

All studies recruited participants from either hospital outpatient or inpatient medical facilities, with none conducted in community settings. Four studies (21.1%) included participants with stroke [26,30,34,38]; nine (47.4%) with heart failure [24,31–33,35–37,39,40]; three (15.8%) with ischaemic heart disease [25,27,28]; and three (15.8%) with ischaemic heart disease or heart failure [29,41,42] (Table 1). A total of 4,397 participants were included, with individual study sizes ranging from 18 to 1,078 participants. The mean age was reported in 14 studies and ranged from 43 to 69 years. The proportion by sex was reported across 15 studies and female sex ranged from 5.5% to 68.4%. Out of 19 studies, five (26.3%) were individual participant level randomised controlled trials [24,30–33], two were cluster randomised trials [34,37], three were non-randomised clinical studies [27,38,41], eight were pre-post intervention studies and one was a retrospective study [25,26,28,29,35,36,39,42].

Only three studies (one evaluating heart failure [24] and two evaluating stroke [26,38]) recruited participants in the acute setting with the remaining involving patients post discharge or community ambulatory patients. Eleven studies (57.9%) were conducted in either tertiary or teaching hospitals [31–39,41,42], one study (5.3%) was conducted across teaching and district-level hospitals [30] and four studies were conducted in rehabilitation centres [25,27–29]. Only one study (5.3%) was specifically conducted in a rural setting [40], and two studies (10.5%) did not report the healthcare facility level [24,26].

## Study quality

The assessment of the quality of studies is summarised in Table 2. Of the 19 studies which were included, five were individual participant level randomised controlled trials [24,30–33]; two were cluster randomised trials [34,37]; three were non-randomised clinical studies [27,38,41]; eight were pre-post intervention studies [25,26,28,29,35,36,39,42]; and one was a retrospective study [40].

All but four studies scored 3–5 out of eight in the quality assessment. One study [40] scored 1 out of 8 and three studies [24,30,34] scored 7 out of 8. The three studies which scored 7 were all randomized controlled trials. Two of these studies [24,34] did not mask both the participants and investigators to treatment allocation. Five studies [25,26,28,29,39] did not report on attrition and one study [33] had follow-up rate of less than 80% (Table 2).

## Types of studies

Across all three cardiovascular pathologies, implementation strategies were targeted at different levels. Of the 19 studies, 14 were targeted at healthcare recipients [25,27–29,31–37,39,41,42], two at healthcare workers [24,40] and three at the healthcare organisation [26,30,38] (Table 3). Most of the heart failure and ischaemic heart studies evaluated interventions targeted at healthcare recipients. Conversely of the four stroke studies, three [26,30,38] evaluated an intervention targeted at the healthcare organisation. Fourteen of the 19 studies evaluated non-pharmacological interventions [25–34,37,38,41,42] with the remaining five implementing an intervention with a pharmacological component [24,35,36,39,40]. All studies evaluating an intervention with a pharmacological component were targeted at populations with heart failure (Table 3).

Five studies, all evaluating heart failure care, tested the role of task sharing with nurse- or pharmacy-led care delivery [35–37,39,40]. Two studies, both evaluating stroke, tested the role of mobile health [30,34]. Twelve studies evaluated care models [24–29,31–33,38,41,42]. Of these, nine were specific to cardiac rehabilitation in the context of heart failure or ischaemic heart disease [25,27–29,31–33,41,42]. The remaining three studies evaluated patients in the acute setting with

**Table 1. Baseline characteristics of included studies.**

| Author, Year, Country | Design and setting | Population | Intervention | Comparator | Sample size | Mean age (years) | Female (%) | Outcome measure(s) | Measurement time points | Effects of the intervention |
|---|---|---|---|---|---|---|---|---|---|---|
| **Heart Failure** | | | | | | | | | | |
| Anane et al, 2013 [35]; Ghana | Pre-post intervention study; Teaching hospital | Adult discharged with HF | Pharmacist led counselling sessions involving education on medical risk management, health behaviour change, and medication compliance. | No | 583 | NR | NR | Functional improvement; all-cause mortality and all cause re- hospitalization | 6 months | New York Heart Association class improved before and after intervention. |
| Eberly et al, 2019 [40]; Rwanda | Retrospective cohort; Rural district level hospital | Adult discharged with HF | Nurses-led delivery care model for heart failure management with monthly supervision from cardiologist. | NR | 719 | NR | 72 | All-cause mortality | 5 years | Improvement in 5-year mortality over time: 38.8% in the first 5-year period (2006–2011) and 27.1% in the second 5-year period (2012–2017) |
| Wondesen A et al, 2022 [36]; Ethiopia | Pre-post intervention study; Tertiary care hospital | Ambulatory adults with HF | Pharmacist and nurse led care providing information and educational material on self-care, physical activity, and medication adherence. Pharmacists identified and resolved drug therapy problems. | No | 412 | 45 | 54 | a) Drug therapy problem (adverse drug reactions, unnecessary drug therapy, need for additional drug therapy, dosage review); b) Medication adherence; and c) Treatment satisfaction | 6 months | Reduced drug therapy problems, improved medication adherence, and increased treatment satisfaction. |
| Ahmed et al., 2021 [39]; Sudan | Pre-post intervention study; Tertiary care hospital | Ambulatory adults with HF | Pharmacist led management of heart failure including initiation, up titration, and changes between drug classes. | No | 110 | 56 | 43 | Achievement of target doses for heart failure pharmacotherapy | 6 months | Improvement in the proportion of patients achieving target doses of guideline directed medical therapy and improvement in LVEF. |
| Dessie et al., 2021 [37]; Ethiopia | Cluster randomized control trial; Tertiary care hospital | Ambulatory adults with HF | Nurse led heart failure self-care educational program comprising of intensive four-day training followed by one-day sessions offered every four months. | Yes, Control group received usual care | 219 | NR | 48 (intervention) and 68 (control) | Hear failure self-care adherence measured | Baseline, 4, 8 and 12 months | Higher self-care adherence scores observed in the intervention group after two and three rounds of educational sessions. |
| Mebazaa et al., 2022 [24]; Mozambique, Nigeria, South Africa | Randomised controlled trial; Setting not reported | Adults with acute HF not on optimal medical therapy. | Risk stratification and post-discharge care based of optimisation of heart failure therapies and in vitro diagnostics | Yes, Control group received usual care | 1078 | 63 | 39 | Heart failure readmission or all cause death | 180 days | Reduction in heart failure readmission or all cause death (15.2% vs. 23.3%, RRR 0.66 [95%CI 0.5 to 0.86], p=0.0021) |

*(Continued)*

**Table 1.** (Continued)

| Author, Year, Country | Design and setting | Population | Intervention | Comparator | Sample size | Mean age (years) | Female (%) | Outcome measure(s) | Measurement time points | Effects of the intervention |
|---|---|---|---|---|---|---|---|---|---|---|
| Ajiboye et al, 2015 [33]; Nigeria | Randomized controlled trial; Teaching hospital | Ambulatory adults with HF | Supervised exercise training including a combination of aerobic and resistance exercises, performed three times a week for 12 weeks. | Yes, Control group received usual care | 51 | 54 | NR | Functional improvement | 12 weeks | Improvement in functional status, haemodynamics and body mass index in the intervention group compared to the control group |
| Awotidebe et al., 2016 [31]; Nigeria | Randomized controlled trial; Teaching hospital | Ambulatory adults with HF | Supervised cardiac rehabilitation exercises that included both aerobic and resistance exercises that increased in intensity weekly. | Yes, Control group randomly assigned to usual pharmacological care | 70 | 69 (intervention) and 64 (control) | 49 (intervention) and 60 control | Functional improvement | 8 weeks | Improvements in functional status using activity of daily living questionnaire in the intervention group compared to the control group |
| Ajiboye et al., 2013 [32]; Nigeria | Randomized controlled trial; Teaching hospital | Ambulatory adults with HF | Supervised cardiac rehabilitation including a combination of aerobic and resistance exercises | Yes, Control group received education and counselling sessions but not exercise training | 38 | 54 | 47 | Haemodynamics | 12 weeks | Improvement in haemodynamics and respiratory parameters |
| **Heart failure/ Ischaemic heart disease** | | | | | | | | | | |
| Ngeno et al., 2022 [41]; Kenya | Nonrandomised clinical study; Teaching hospital | Ambulatory adults with HF including those with cardiac ischaemia | Non randomly enrolled to institution-based supervised cardiac rehabilitation, home based cardiac rehabilitation or an observation group. | Yes, enrolled to observation arm/control group | 100 | 51 | 28 | Protocol adherence along with functional improvement | 1 month | Higher protocol adherence observed in the institution-based rehabilitation group. All three arms showed improvement in functional status. |
| Digenio A.G et al., 1996 [29]; South Africa | Pre-post intervention study; Rehabilitation Centre | Patients discharged following an acute myocardial infarction and documented left ventricular impairment | Medically supervised exercise training programme | No | 28 | 64 | NR | Haemodynamics, left ventricular function and effort tolerance | 6 months | Improvement in exercise capacity. No change in left ventricular function at rest or during exercise. |
| Kpadonou et al 2013 [42]; Benin | Pre-post intervention study; Teaching hospital | Ambulatory patients diagnosed with HF, coronary disease, or hypertension. | Institution based supervised exercise training | No | 27 | 46 (Coronary disease); 43 (HF); 43 (Hypertension) | 26 | Haemodynamics | 10 weeks | Improvement in haemodynamics |

*(Continued)*

**Table 1.** (Continued)

| Author, Year, Country | Design and setting | Population | Intervention | Comparator | Sample size | Mean age (years) | Female (%) | Outcome measure(s) | Measurement time points | Effects of the intervention |
|---|---|---|---|---|---|---|---|---|---|---|
| **Ischaemic heart disease** | | | | | | | | | | |
| van Rooy & Y Coopoo, 2017 [25]; South Africa | Pre-post intervention study; Rehabilitation Centre | Ambulatory patients who had undergone coronary artery bypass grafting | Supervised Individualized exercise programmes and provision of lifestyle manual | No | 18 | NR | 6 | Cardiovascular disease risk; change in lifestyle habits, nutrition knowledge and physical activity profile | 12 weeks | The intervention reduced all evaluated indices of cardiovascular disease risk and increased physical activity levels |
| Morris et al., 1993 [28]; South Africa | Pre-post intervention study; Rehabilitation Centre | Ambulatory patients with ischaemic heart disease | Cardiac rehabilitation programme consisting of aerobic exercise sessions and advice on lifestyle modification. | No | 108 | 57 | 12 | Change in lipid profile and exercise capacity | 6 months | Improvement in exercise capacity and lipid profile following cardiac rehabilitation |
| Joughin et al., 1999 [27]; South Africa | Non randomised clinical study; Rehabilitation Centre | Ambulatory patients with coronary heart disease | Endurance training including combination of walking and jogging on a measured outdoor circuit or cycling on a stationary ergometer. Compliers allocated to intervention group were compared to non- compliers/ dropouts allocated to control group | Yes, Patient dropping out of cardiac rehabilitation assigned to control group. | 111 | 57 (intervention); 59 (control) | 9 (intervention); 6 (control) | Haemodynamics and exertional tolerance | 6month and 18month after intervention | Improvement in haemodynamics and exercise capacity. |
| **Stroke** | | | | | | | | | | |
| Nakibuuka et al., 2016 [38]; Uganda | Non randomised clinical study; Tertiary care hospital | Hospitalised patients with stroke | Stroke care bundle including rapid initial stroke screening; brain imaging; bedside swallow evaluation; aspirin administration; physiological monitoring and management; and early rehabilitation | Yes, control group received usual care | 254 | NR | NR | Mortality and functional improvement | 30 days post stroke | No improvement in mortality rates or functional outcomes following implementation of stroke care bundle. |
| Sarfo et al., 2019 [34]; Ghana | Cluster randomized controlled trial; Teaching hospital | Ambulatory patients with recent stroke and uncontrolled hypertension | Mobile health nurse guided intensive home blood pressure monitoring with the use of bluetooth blood pressure device and smartphone. | Yes, Randomized control selection, unmasked, control received text message on healthy lifestyle behaviour but not medication adherence | 60 | 54 (intervention); 56 (control) | 40 (intervention); 30 (control) | Blood pressure control | 9 months | Study showed feasibility and signal of improvement in blood pressure control. |

*(Continued)*

**Table 1.** (Continued)

| Author, Year, Country | Design and setting | Population | Intervention | Comparator | Sample size | Mean age (years) | Female (%) | Outcome measure(s) | Measurement time points | Effects of the intervention |
|---|---|---|---|---|---|---|---|---|---|---|
| Owolabi et al., 2019 [30]; Nigeria | Randomized controlled trial; Teaching and district hospitals | Ambulatory patients with recent stroke within the last year. | Chronic care model components of care delivery system redesign (follow-up visits and pre-appointment phone texts); self-management support; and clinical information systems (e.g., hospital registry). | Yes, Controls received usual care with name and contact information of a phone contact | 400 | 57 | 37 | Blood pressure control | 12 months | No significant difference in systolic blood pressure reduction between the two groups. |
| De Villiers et al., 2009 [26]; South Africa | Pre-post intervention study; Hospital setting | Hospitalised patients with a clinical diagnosis of stroke | Stroke unit implementation delivering multidisciplinary stroke care including a) stroke treatment guidelines and b) twice weekly stroke ward rounds with allied health professionals | No | 195 | 59 | 60 | a) Length of hospital stay; b) in-hospital mortality; c) transfer to referral hospital; d) number of patients who had CT brain scans performed | 3 months pre- and post-intervention | Decrease in-hospital mortality (33% to 16%) and increase in hospital stay from 5 to 7 days; increase in referral to inpatient rehabilitation (5% to 19%). No increase in number of CT brain scans and number of referrals to tertiary facility |

two evaluating care models in stroke [26,38]; one in heart failure [24] and one evaluating multidisciplinary medical models in stroke [30]. These studies are discussed in detail, by cardiovascular pathology, below.

## Heart failure

Twelve studies recruited patients with heart failure. Among these, nine studies (75%) specifically recruited patients with heart failure [24,31–33,35–37,39,40] and three studies included patients with heart failure or ischaemic heart disease [29,41,42]. Only one study [24] targeted care delivery of acute heart failure, with the remaining designed to improve care for recently discharged or ambulatory patients with heart failure. Ten of the 12 studies (83%) targeted an intervention at the healthcare recipient [29,31–33,35–37,39,41,42], two targeted the healthcare worker [24,40] and none targeted the healthcare organisation. Of the healthcare recipient targeted interventions, three addressed medication compliance and up-titration in chronic heart failure pharmacotherapy [35,36,39]. These included pharmacy and nurse-led interventions to target appropriate up-titration and ensure compliance of guideline directed medical therapy in ambulatory patients [35,36,39]. These studies showed either improved adherence or treatment satisfaction alongside an increase in the proportion of patients receiving goal directed medical therapy. One of these studies provided medical counselling to hospitalised patients with heart failure in Ghana, showing better clinical outcomes at 6 month follow up [35].

Implementation of supervised cardiac rehabilitation care models was the focus of six studies [29,31–33,41,42] with five implementing institution-based exercise programmes [29,31–33,42] and one study from Kenya [41] comparing home-based with institution-based cardiac rehabilitation. All six studies reported improvements across a variety of physiological

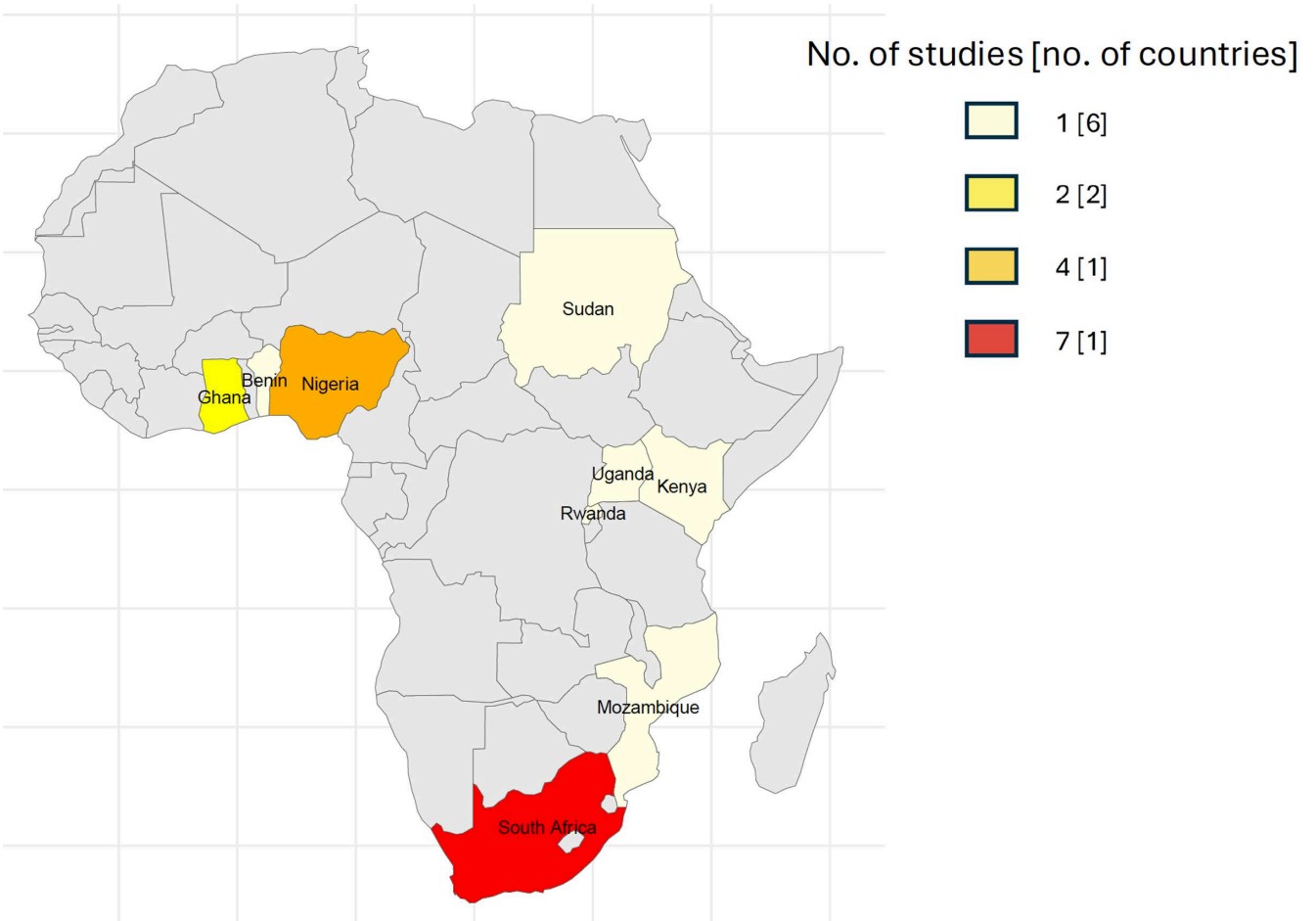

**Fig 2. Cartogram highlighting countries contributing studies to this systematic review.** The colours illustrate the numerical scale for the number of studies. The colour box in the legend summarises the number of studies with the number in the squared brackets summarising the number of countries. Figure created in R Version 4.4.0 using the rnaturalearth package. Source of base map is from natural earth available at http://www.naturalearthdata.com/about/terms-of-use/ (accessed 8th April 2025).

measures. Educational interventions were also prominent, with three of the interventions targeted at healthcare recipients, providing education about heart failure and support health-promoting behaviours [35–37].

STRONG-HF, a large multinational randomised controlled trial recruited heart failure patients in South Africa, Mozambique, and Nigeria. The study implemented a care model which incorporated an intensive follow-up regimen post discharge for patients who had been admitted with acute heart failure [24]. The regimen was based on risk-stratification using circulating biomarkers of cardiac remodelling, and showed a significant improvement in a composite of all-cause mortality and heart failure rehospitalization.

**Stroke**

Studies recruiting patients with stroke evaluated implementation strategies in both the inpatient [26,38] and outpatient hospital settings [30,34]. In the inpatient setting, two studies implemented structured stroke care. De Villiers *et al.* established a multi-disciplinary stroke unit within the medical ward engaging allied healthcare professionals and

**Table 2. Quality assessment of included studies.**

| Reference and country | Design | cohort | control or comparison group | pre/post intervention data | random assignment of participants to the intervention | random selection of participants for assessment | follow up rate of >80% | comparison groups equivalent on sociodemographic | comparison groups equivalent at baseline on outcome measures | Score |
|---|---|---|---|---|---|---|---|---|---|---|
| Nakibuuka et al. 2016 [38]; Uganda | Nonrandomised clinical study | yes | yes | na | no | yes | yes | yes | na | 5 |
| Ahmed et al. 2021 [39]; Sudan | Pre-post intervention study | yes | no | yes | no | no | nr | na | na | 2 |
| Wondesen A et al 2022 [36]; Ethiopia | Pre-post intervention study | yes | no | yes | yes | yes | yes | yes | na | 6 |
| Dessie et al 2021 [37]; Ethiopia | Cluster randomised control trial | yes | yes | yes | yes | no | yes | no | yes | 6 |
| Sarfo et al 2019 [34]; Ghana | Cluster randomised control trial | yes | yes | yes | yes | no | yes | yes | yes | 7 |
| Anane et al 2013 [35]; Ghana | Pre-post intervention study | yes | no | yes | no | no | yes | na | na | 3 |
| Ngeno et al 2022 [41]; Kenya | Nonrandomised clinical study | yes | yes | no | no | no | yes | no | no | 3 |
| Eberly et al 2019 [40]; Rwanda | Retrospective review of intervention | yes | na | no | na | no | na | nr | nr | 1 |
| Ajiboye et al 2015 [33]; Nigeria | Randomised controlled trial | yes | yes | no | yes | no | no | yes | yes | 5 |
| Owolabi et al 2019 [30]; Nigeria | Randomised controlled trial | yes | yes | yes | yes | no | yes | yes | yes | 7 |
| de Villiers et al 2009 [26]; South Africa | Pre-post intervention study | yes | no | yes | no | no | nr | no | nr | 2 |
| Van Rooy & Y Coopoo 2017 [25]; South Africa | Pre-post intervention study | yes | no | yes | no | no | nr | na | na | 2 |
| Awotidebe et al. 2016 [31]; Nigeria | Randomised controlled trial | yes | yes | no | yes | no | yes | no | yes | 5 |
| Ajiboye et al 2013 [32]; Nigeria | Randomised controlled trial | yes | yes | no | yes | no | yes | yes | yes | 6 |

*(Continued)*

**Table 2.** (Continued)

| Reference and country | Design | cohort | control or comparison group | pre/post intervention data | random assignment of participants to the intervention | random selection of participants for assessment | follow up rate of >80% | comparison groups equivalent on sociodemographic | comparison groups equivalent at baseline on outcome measures | Score |
|---|---|---|---|---|---|---|---|---|---|---|
| Kpadonou et al 2013 [42]; Benin | Pre-post intervention study | yes | no | yes | no | no | yes | na | na | 2 |
| Joughin et al 1999 [27]; South Africa | Non randomised clinical study | yes | yes | yes | no | no | yes | no | no | 4 |
| Digenio et al 1996 [29]; South Africa | Pre-post intervention study | yes | no | yes | no | no | nr | na | na | 2 |
| Morris et al 1993 [28]; South Africa | Pre-post intervention study | yes | no | yes | no | no | nr | na | na | 2 |
| Mebazaa et al 2022 [24]; Mozambique, Nigeria, South Africa | Randomised controlled trial | yes | yes | yes | yes | no | yes | yes | yes | 7 |

creating protocols to enhance stroke care [26]. This approach reduced inpatient mortality and increased referrals to rehabilitation centres. Conversely, Nakibuuka *et al.* implemented a stroke care bundle at a tertiary hospital in Uganda for acute stroke patients [38]. A formal acute stroke care pathway was implemented which included haemodynamic and glycaemic monitoring; brain imaging; and administration of pharmacotherapy. Funds were made available to implement the care pathway. Importantly, the acute care pathway was implemented for a period of 72 hours, followed by usual care. However, the study did not show any improvement in patient mortality following implementation. This may reflect differences in patient characteristics including greater stroke severity in patients enrolled during the intervention phase.

In the outpatient setting, two studies attempted to improve blood pressure control in stroke survivors [30,34]. The PINGS trials [34] implemented nurse-guided intensive home blood pressure monitoring facilitated by a smartphone application, while the THRIVES trial [30] employed text-message reminders, tailored disease self-management report cards, and educational videos. Both these interventions showed feasibility. The THRIVES trial did not demonstrate a significant improvement in blood pressure control, whilst the PINGS feasibility study did show a signal towards improved blood pressure control.

### Ischaemic heart disease

Three studies [25,27,28] solely evaluated patients with ischaemic heart disease. All studies looked at cardiac rehabilitation care models and originated from South Africa, showing an improvement in haemodynamics and parameters of exertional tolerance. Our systematic review did not find any studies evaluating care models for patients with ischaemic heart disease in the acute setting including those presenting with acute coronary syndrome.

### Discussion

Our systematic review summarises the body of literature evaluating implementation strategies of evidence-based interventions for management of patients with CVD in SSA. We identified 19, of which five were individual participant level

**Table 3. Summary of implementation attributes for each study.**

| Author, Year, Country | Intervention type | Intervention strategy target |
|---|---|---|
| **Heart Failure** | | |
| Anane et al, 2013 [35]; Ghana | Pharmacological and non-pharmacological | Healthcare recipients |
| Eberly et al, 2019 [40]; Rwanda | Pharmacological and non-pharmacological | Healthcare recipients and workers |
| Wondesen A et al, 2022 [36]; Ethiopia | Pharmacological and non-pharmacological | Healthcare recipients |
| Ahmed et al., 2021 [39]; Sudan | Pharmacological and non-pharmacological | Healthcare recipients |
| Dessie et al., 2021 [37]; Ethiopia | Non-pharmacological | Healthcare recipients |
| Mebazaa et al., 2022 [24]; Mozambique, Nigeria, South Africa | Pharmacological | Healthcare workers |
| Ajiboye et al, 2015 [33]; Nigeria | Non-pharmacological | Healthcare recipients |
| Awotidebe et al., 2016 [31]; Nigeria | Non-pharmacological | Healthcare recipients |
| Ajiboye et al., 2013 [32]; Nigeria | Non-pharmacological | Healthcare recipients |
| **Heart failure/ Ischaemic heart disease** | | |
| Ngeno et al., 2022 [41]; Kenya | Non-pharmacological | Healthcare recipients |
| Digenio A.G et al., 1996 [29]; South Africa | Non-pharmacological | Healthcare recipients |
| Kpadonou et al, 2013 [42]; Benin | Non-pharmacological | Healthcare recipients |
| Ischaemic heart disease | | |
| van Rooy & Y Coopoo, 2017 [25]; South Africa | Non-pharmacological | Healthcare recipients |
| Morris et al., 1993 [28]; South Africa | Non-pharmacological | Healthcare recipients |
| Joughin et al., 1999 [27]; South Africa | Non-pharmacological | Healthcare recipients |
| **Stroke** | | |
| Nakibuuka et al., 2016 [38]; Uganda | Non-pharmacological | Healthcare organisation |
| Sarfo et al., 2019 [34]; Ghana | Non-pharmacological | Healthcare recipients |
| Owolabi et al., 2019 [30]; Nigeria | Non-pharmacological | Healthcare organisation |
| De Villiers et al., 2009 [26]; South Africa | Non-pharmacological | Healthcare organisation |

randomised controlled trials [24,30–33]; two were cluster randomised trials [34,37]; three were non-randomised clinical studies [27,38,41]; eight were pre-post intervention studies [25,26,28,29,35,36,39,42]; and one was a retrospective study [40]. Of the seven randomised trials, four evaluated cardiac rehabilitation care models in the context of heart failure, showing improvements in exertional tolerance or cardiac haemodynamics [31–33,37]. Mebazaa et al showed a significant reduction in hospitalisation and mortality, following randomisation to biomarker-guided post discharge care [24]. The remaining two randomised trials [30,34] were in patients with stroke using text messages to improve blood pressure control. Both showed feasibility with one study showing a signal towards better blood pressure control. Of the eight pre-post intervention studies, four were based on cardiac rehabilitation in patients with heart failure or ischemic heart disease [25,28,29,42]. Similar to randomised trials [31–33,37], the majority of these showed improvement in exertional tolerance or haemodynamics. Two pre-post intervention studies were pharmacist led, showing improved medical adherence and target drug dose achievement [36,39].

Several important observations emerge from our review. First, most studies did not focus on patients in the acute or peri-discharge period. This is important, as the highest risk of morbidity and mortality is experienced by patients during this period, either as inpatients or shortly after discharge [43]. Second, studies specifically targeting implementation strategies of pharmacotherapy, diagnostic tools, multi-disciplinary teams or risk stratification approaches in the acute

setting were scarce; we found only one study for acute heart failure and two for stroke. We found none evaluating patients with acute coronary syndrome. Third, only three studies [26,30,40] recruited patients from non-teaching or non-tertiary care settings. This finding has particular relevance to the region, given that nearly 6 out 10 people in SSA do not reside in urban regions [44]. The applicability of study findings from urban settings to rural ones will further be hindered given differences in health infrastructure, healthcare provider expertise and availability of CVD care at lower-level healthcare facilities. Fourth, the studies that we identified only covered 10 of the 49 countries in the SSA. This limits generalizability of our study findings to the SSA region and further demonstrate substantial evidence gaps in CVD care implementation in the region. Finally, it was encouraging to observe that many studies did evaluate the implementation of cardiovascular rehabilitation programmes, an area that is generally given less importance.

Despite SSA bearing a significant and increasing global burden of CVD, its populations remain underrepresented in both evidence generation and implementation. The prognosis following acute cardiac pathologies in SSA remains unacceptably poor, with a quarter of all adult hospitalizations in certain countries being cardiovascular-related. Despite being a decade younger on average compared to high-income countries, cardiac patients in SSA experience up to 3-fold higher short-term case fatality rates (30% vs. 9%) [9,45]. Annually, low and middle-income countries witness an estimated 16 million excess deaths, with over a third attributed to CVD. Approximately 4.5 million of these deaths occur in the African region, predominantly due to inadequate secondary care health systems [46]. These deficiencies contribute as significantly to excess deaths as immature primary care systems in Africa. The high mortality, at least in part, may be exacerbated by poor links between different levels of care. Furthermore, despite the high burden of CVD and poor outcomes, research funding for the discipline has been disproportionately low.

Provision of appropriate pharmacotherapy across SSA patients with established CVD remains poor. Across SSA, only 1 in 10 patients with established CVD use aspirin [16], and less than 50% of heart failure patients receive appropriate management in Kenya [47]. Low uptake of appropriate pharmacotherapeutic and non-pharmacotherapeutic approaches across patients in SSA with established CVD is likely to be due to inadequacies across several implementation strategic domains [48]. These implementation strategies specifically tackle the 'how to' component for delivering optimal clinical care with recent guidance published on defining and operationalising these strategies [17,18]. These guidelines should be considered by implementation researchers and stakeholders aiming to improve care in patients with acute or established CVD in SSA.

The lack of published literature, especially in the acute cardiovascular care setting, may reflect the current funding priorities in global cardiovascular health. For example, from 2017-2024, UK funding bodies, like the NIHR, allocated over £370 million in healthcare research, with less than 1% (~£2 million) directed towards management of patients with acute or established CVD compared to £40 million for primary cardiovascular care [49]. The mismatch between disease burden due to established cardiovascular disease and research funding underscores the importance of further research in the area. Efforts now need to concentrate on the optimal approaches in implementing evidence-based therapy for the three core cardiovascular pathologies. These efforts need to be contextualised to the settings, considering the differing healthcare infrastructure both within and between countries across the African continent.

Several limitations of our work should be considered. The literature search was confined to four databases and excluded grey literature, potentially omitting relevant unpublished studies. Furthermore, our focus was restricted to specific cardiovascular pathologies, namely stroke, heart failure and ischaemic heart disease. Whilst these pathologies constitute most of CVD, our review does not shed any light on gaps in the implementation strategies of other prevalent CVD such as peripheral vascular disease.

## Recommendations

Our review highlights several areas where the findings from implementation studies may improve outcomes in patients with acute or established CVD and point towards areas of further research. There is relatively strong evidence for

implementation of cardiac rehabilitation programmes, particularly in the context of heart failure and ischaemic heart disease. Implementation of these programmes, if contextualised to rural settings, can improve functional and clinical outcomes. Our evidence synthesis highlights that the implementation of acute care bundles has mainly been tested in the field of stroke medicine. Urgent evidence is now needed in the acute cardiovascular setting, particularly in the context of diagnosing and treating acute heart failure and acute coronary syndrome.

## Conclusion

This evidence synthesis highlights significant gaps in the implementation strategies for managing acute or established CVD in SSA. Particular gaps were highlighted in the acute care setting, specifically related to acute coronary syndrome and implementation strategies targeting pharmacotherapeutic optimizations. We also highlight a notable lack of studies focusing on effective implementation strategies in primary care facilities and lower-level hospital settings.

## Supporting information

**S1 Checklist. PRISMA checklist for the systematic review.**
(DOCX)

**S1 Table. Details of the search criteria stratified by database.**
(XLSX)

**S1 Text. List of hits from the initial abstract and title screening alongside list of full papers reviewed and selected for the systematic review.**
(DOCX)

## Author contributions

**Conceptualization:** Leah A Sanga, Jonathan A Hudson, Anoop S V Shah.

**Formal analysis:** Jonathan A Hudson.

**Funding acquisition:** Pablo Perel.

**Methodology:** Leah A Sanga, Jonathan A Hudson, Anoop S V Shah.

**Project administration:** Leah A Sanga, Pablo Perel, Anoop S V Shah.

**Resources:** Jonathan A Hudson.

**Supervision:** Pablo Perel, Anoop S V Shah.

**Writing – original draft:** Leah A Sanga, Jonathan A Hudson, Anoop S V Shah.

**Writing – review & editing:** Jonathan A Hudson, Alexander D Perkins, Adrianna Murphy, Anthony Etyang, Pablo Perel, Anoop S V Shah.

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
