## [Decision Letter · Decision Letter 0]

16 Sep 2024

PGPH-D-24-01693

Implementation strategies to improve outcomes in patients with established cardiovascular disease in sub-Saharan Africa: a systematic review

Dear Shah,

Thank you for submitting your manuscript to PLOS Global Public Health. After careful consideration, we feel that it has merit but does not fully meet PLOS Global Public Health’s publication criteria as it currently stands. Therefore, we invite you to submit a revised version of the manuscript that addresses the points raised during the review process.

We look forward to receiving your revised manuscript.

Kind regards,

Collins Otieno Asweto, PhD

Academic Editor

Journal Requirements:

1. Your current Financial Disclosure states, “NIHR”. However, your funding information on the submission form indicates that you received funding from “National Institute for Health and Care Research” and Grant Recipient "Pablo Perel". Please indicate by return email the full and correct funding information for your study and confirm the order in which funding contributions should appear. Please be sure to indicate whether the funders played any role in the study design, data collection and analysis, decision to publish, or preparation of the manuscript.

2. We ask that a manuscript source file is provided at Revision. Please upload your manuscript file as a .doc, .docx, .rtf or .tex.

3. Please provide separate figure files in .tif or .eps format.

4. We notice that your supplementary "Supplementary material-Search criteria" are included in the manuscript file. Please remove them and upload them with the file type 'Supporting Information'. Please ensure that each Supporting Information file has a legend listed in the manuscript after the references list.

5. Please provide a complete Data Availability Statement in the submission form, ensuring you include all necessary access information or a reason for why you are unable to make your data freely accessible. If your research concerns only data provided within your submission, please write "All data are in the manuscript and/or supporting information files" as your Data Availability Statement.

6. Figure 2: please (a) provide a direct link to the base layer of the map (i.e., the country or region border shape) and ensure this is also included in the figure legend; and (b) provide a link to the terms of use / license information for the base layer image or shapefile. We cannot publish proprietary or copyrighted maps (e.g. Google Maps, Mapquest) and the terms of use for your map base layer must be compatible with our CC-BY 4.0 license. 

Additional Editor Comments (if provided):

Reviewers' comments:

Reviewer's Responses to Questions

**Comments to the Author**

1. Does this manuscript meet PLOS Global Public Health’s publication criteria ? Is the manuscript technically sound, and do the data support the conclusions? The manuscript must describe methodologically and ethically rigorous research with conclusions that are appropriately drawn based on the data presented.

Reviewer #1: Yes

Reviewer #2: Partly

Reviewer #3: Yes

2. Has the statistical analysis been performed appropriately and rigorously?

Reviewer #1: N/A

Reviewer #2: I don't know

Reviewer #3: N/A

3. Have the authors made all data underlying the findings in their manuscript fully available (please refer to the Data Availability Statement at the start of the manuscript PDF file)?

Reviewer #1: Yes

Reviewer #2: Yes

Reviewer #3: Yes

4. Is the manuscript presented in an intelligible fashion and written in standard English?

Reviewer #1: Yes

Reviewer #2: No

Reviewer #3: Yes

5. Review Comments to the Author

Reviewer #1: The main objective of the research article was to describe, synthesize and identify key gaps in

the implementation strategies of evidence-based approaches that improves clinical outcomes for patients with CVD in SSA.

The strength of the manuscript is to provide helpful insights about gaps in implementation strategies that improve care outcomes of CVD patients in SSA. The research article describes extremely important health topic of interest, contributing to preventable mortality from CVDs creating substantial tall of already strained health systems of SSA countries. The article calls for increased investments, including in in interventions and research targeting care of patient with CVD to achieve SDGs.

The actionability of the manuscript could be improved by standardizing description of interventions and better systematizing implementation strategies by different domains. This will help to differentiate clinical interventions from various implementation strategies from one hand and identifying explicit gaps in implementation strategies from another to inform programmatic and learning agenda.

Specific recommendations are detailed below.

The lines of the manuscript are not numbered per PLOS LaTeX template. In the review, I will therefore highlight the feedback per sections without indicating line numbers.

Exclusion criteria

• It may be helpful to add primary prevention of CVD as an exclusion criterion, considering that study explicitly focuses on acute or established CVD. If this criterion was considered at a later stage (during expert consensus or consultation with 3rd reviewer or during the read of preselected 34 studies), please, state so.

Table 1

• To better identify and systematize gap in implementation strategies, it would be helpful to 1st classify evidence-based interventions for management of patients with CVD beyond pharmacologic/non-pharmacologic.

• Similarly, implementation strategies could be further systematized by interventions targeting patient or families (demand side); improving point-of care service provision (point of care interventions) and broader, above site, health system strengthening interventions.

• When systematizing the interventions by various implementation strategies (e.g. task sharing, model of care), it would be helpful to refer to any specific systematization used for these purposes noting key domains used to systematize interventions. For example, the research article sites EPIC implementation strategy paper in the discussion, which provides compiled list of strategies to implement change in clinical settings (https://implementationscience.biomedcentral.com/articles/10.1186/s13012-015-0209-1/tables/3). There may be other, more relevant systematization of implementation strategies that could be used to better describe interventions. This will help to better differentiate clinical interventions from various implementation strategies from one hand and identifying gaps in implementation strategies from another.

• Suggest using referenced article (Proctor, E.K., Powell, B.J. & McMillen, J.C. Implementation strategies: recommendations for specifying and reporting. Implementation Sci 8, 139 (2013). https://doi.org/10.1186/1748-5908-8-139) to specify each implementation strategy, described in the table 1

Discussion

• Page 2 states that “only three studies [25,29,39] recruited patients from non-teaching or non-tertiary care settings. This finding has particular relevance to the region, given that nearly 6 out 10 people in SSA do not reside in urban regions.[43]” Another important reason for this could be limited availability of CVD care in lower-level health facilities in SSA.

• The fact that studies covered only 10 out of over 50 SSA countries limit generalizability of the study findings to entire SSA countries and further demonstrate substantial evidence gap in CVD care implementation best practices in the region. I suggest to explicitly state these in relevant sections of the article.

Conclusion

• Gaps in implementation strategies related to evaluating diagnostics could be partly due to limiting the systematic review to already diagnosed CVDs. Suggest removing “evaluating diagnostics” from the conclusion unless you have a strong justification.

• Suggest revising sentence” Particular gaps were highlighted in the acute care setting, specifically evaluating diagnostics and pharmacotherapeutic optimizations” as “Particular gaps were highlighted in the acute care setting, specifically related to acute coronary syndrome and implementation strategies targeting pharmacotherapeutic optimizations.”

• As noted earlier, limited number of studies in non-tertiary and teaching hospital settings may be related to poor availability of CVD care in lower-level health facilities. I therefore suggest reviewing the sentence: “We also highlight a notable lack of studies focusing on effective implementation strategies in non-urban settings” as “We also highlight a notable lack of studies focusing on effective implementation strategies in primary care facilities and lower-level hospital settings”.

Reviewer #2: PLOS Global Public Health

Implementation strategies to improve outcomes in patients with established cardiovascular disease in sub-Saharan Africa: a systematic review

--Manuscript Draft--

The authors have presented an insightful research with a concise and clear title. The research has a good contextual background with relevant research objectives. However, the authors need to pay attention to some of the highlighted observations in the manuscript presented. There were numerous typographical errors, inconsistencies in writings and representations. The references were improperly cited with inconsistent referencing style. The tables appear cumbersome with some fragmented words. While reading through the manuscript, there was a noticeable disconnect between the objectives, results and discussion. The page numbering was not serialized; the authors should carefully address all these concerns. ( Check comments on the attachment).

I commend the authors for this meta-analysis study which truly will bridge a gap in knowledge not only in Sub-Saharan Africa but worldwide.

Reviewer #3: This systematic review aims to synthesise and highlight critical gaps in the implementation of evidence-based cardiovascular disease (CVD) interventions in Sub-Saharan Africa. The manuscript has several strengths, including adherence to PRISMA guidelines, a comprehensive database search, clear eligibility criteria, and the use of the Evidence Project risk of bias tool. The review identifies significant gaps in the implementation of CVD interventions, particularly in acute care settings and for hospitalized patients. While the authors are off to a good start, they should provide more details to enhance the methodological rigor of the review. Additionally, I recommend a thorough proofreading of the manuscript for editorial issues, including formatting the tables, enhancing clarity, and spelling out numbers below ten.

Methods:

1. The authors state that the study was reported in accordance with the PRISMA guidelines (Page 6, Data source and search strategy). However, the PRISMA checklist is not included in the manuscript or supplementary file.

2. The authors should indicate if the review protocol was pre-registered in PROSPERO.

3. The authors mention that the bibliography of key papers was searched (Page 6, Data source and search strategy), but it is unclear what these key papers are and how they were identified.

4. The authors mention that they did not restrict their search by language (Page 6, Data source and search strategy). It would be helpful to clarify whether any of the screened studies were in languages other than the primary language of the review and how such studies were managed, including any translation processes or resources used to ensure accurate data extraction and analysis.

5. The authors state that CVD was restricted to recurrent myocardial infarction, stroke, heart failure, and ischemic heart disease (Page 6, inclusion criteria), but it is unclear why other forms of CVD (such as peripheral artery disease) were excluded.

6. While the authors use the Evidence Project risk of bias tool (Page 7/8, evidence synthesis and risk of bias), it would be helpful to explain why this specific tool was chosen over others.

7. Although the description of the risk of bias tool's criteria is clear, there is no information on how the scores were interpreted or what thresholds were used to categorise studies as having high or low risk of bias.

6. PLOS authors have the option to publish the peer review history of their article (what does this mean? ). If published, this will include your full peer review and any attached files.

**Do you want your identity to be public for this peer review?** For information about this choice, including consent withdrawal, please see our Privacy Policy .

Reviewer #1: **Yes: ** Tamar Chitashvili

Reviewer #2: **Yes: ** PRISCILIA UHUANMWEN IMADE

Reviewer #3: No

---

## [Decision Letter · Decision Letter 1]

16 Dec 2024

PGPH-D-24-01693R1

Implementation strategies to improve outcomes in patients with established cardiovascular disease in sub-Saharan Africa: a systematic review

Dear Dr. Shah,

Thank you for submitting your manuscript to PLOS Global Public Health. After careful consideration, we feel that it has merit but does not fully meet PLOS Global Public Health’s publication criteria as it currently stands. Therefore, we invite you to submit a revised version of the manuscript that addresses the points raised during the review process.

The manuscript has been evaluated by three reviewers, and their comments are available below.

The reviewers are largely happy with the revision, but have requested a few further clarifications to the text, further discussion, and further proofreading.

Could you please revise the manuscript to carefully address the concerns raised?

We look forward to receiving your revised manuscript.

Kind regards,

Helen Howard

Staff Editor

Journal Requirements:

1. As required by our policy on Data Availability, please ensure your manuscript or supplementary information includes the following:

Additional Editor Comments (if provided):

Reviewers' comments:

Reviewer's Responses to Questions

**Comments to the Author**

1. If the authors have adequately addressed your comments raised in a previous round of review and you feel that this manuscript is now acceptable for publication, you may indicate that here to bypass the “Comments to the Author” section, enter your conflict of interest statement in the “Confidential to Editor” section, and submit your "Accept" recommendation.

Reviewer #1: All comments have been addressed

Reviewer #3: All comments have been addressed

Reviewer #4: (No Response)

2. Does this manuscript meet PLOS Global Public Health’s publication criteria ? Is the manuscript technically sound, and do the data support the conclusions? The manuscript must describe methodologically and ethically rigorous research with conclusions that are appropriately drawn based on the data presented.

Reviewer #1: Yes

Reviewer #3: Yes

Reviewer #4: Yes

3. Has the statistical analysis been performed appropriately and rigorously?

Reviewer #1: Yes

Reviewer #3: N/A

Reviewer #4: Yes

4. Have the authors made all data underlying the findings in their manuscript fully available (please refer to the Data Availability Statement at the start of the manuscript PDF file)?

Reviewer #1: Yes

Reviewer #3: Yes

Reviewer #4: Yes

5. Is the manuscript presented in an intelligible fashion and written in standard English?

Reviewer #1: Yes

Reviewer #3: Yes

Reviewer #4: Yes

6. Review Comments to the Author

Reviewer #1: Thank you for taking time and reviewing the manuscript. Changes made truly makes the article clearer and more actionable.

Reviewer #3: The authors have addressed all coments, and I have no further comments. I recommend that the manuscript to be accepted.

Reviewer #4: A good overview and reasonable analysis, given the scarcity of research in this area specific to the SSA countries. The previous reviewer's comments have been mostly addressed, and the conclusions & limitations of this study strengthened.

A few additional comments below :

1. On page 12, under 'Stroke', there is a comment that there was no difference in outcomes between the 'multidisciplinary stroke unit with the medical ward' and the 'stroke care bundle at a tertiary hospital'. It is not clear from this paragraph or Table 1 what was the key difference between these 2 implementation strategies. It would be useful to highlight if there were any key differences (even in the cost of these 2 strategies, for example).

2. While the conclusion understandably highlighted the many gaps, there may have been some studies from which useful intervention strategies could be identified (e.g. the ones highlighted in point 1 above) - and the authors could, perhaps, include a 'what next' or a section on 'recommendations' - e.g. identifying the more promising interventions to ascertain if they can be replicated in more rural settings, as you have mentioned that 60% of the SSA population live in a rural setting. This would make for a more robust paper that doesn't just identify issues but make sound recommendations to move this important issue forward.

3. A few typos still to be corrected - Pg 10 - '"health failure" instead of "heart failure". Pg 19 - para beginning with "Provision of..." - 2nd sentence has "across SSA" twice. Please check rest of document too.

7. PLOS authors have the option to publish the peer review history of their article (what does this mean? ). If published, this will include your full peer review and any attached files.

**Do you want your identity to be public for this peer review?** For information about this choice, including consent withdrawal, please see our Privacy Policy .

Reviewer #1: **Yes: ** Tamar Chitashvili

Reviewer #3: No

Reviewer #4: No

---

## [Editor Report · Decision Letter 2]

2 Apr 2025

Implementation strategies to improve outcomes in patients with established cardiovascular disease in sub-Saharan Africa: a systematic review

PGPH-D-24-01693R2

Dear Dr Shah,

We are pleased to inform you that your manuscript 'Implementation strategies to improve outcomes in patients with established cardiovascular disease in sub-Saharan Africa: a systematic review' has been provisionally accepted for publication in PLOS Global Public Health.

Best regards,

Julia Robinson

Executive Editor